# Manufacturing and Testing of a Variable Chord Extension for Helicopter Rotor Blades

**Christoph Balzarek** [1] , **Steffen Kalow** [1], **Johannes Riemenschneider** [1,*] **and Andres Rivero** [2]

1   Institute of Composite Structures and Adaptive Systems, German Aerospace Center, Lilienthalplatz 7, 38108 Braunschweig, Germany; christoph.balzarek@dlr.de (C.B.); steffen.kalow@dlr.de (S.K.)

2   Research Associate in Morphing Structures, Bristol Composites Institute, University of Bristol, Queen's Building, University Walk, Bristol BS8 1TR, UK; andres.riverobracho@bristol.ac.uk

*   Correspondence: johannes.riemenschneider@dlr.de; Tel.: +49-531-295-2388

**Abstract:** Helicopters are still an indispensable addition to aviation in this day and age. They are characterized by their ability to master both forward flight and hover. These characteristics result in a wide range of possible operations. Key for the design of the rotor blades is a blade design that always represents a compromise between the different flight conditions, which enables safe and efficient flight in the various flight conditions. In order to operate the rotor blade even more efficiently in all flight conditions, a new morphing concept, the so-called linear variable chord extension, has been developed. Here, the blade chord length in the root area is changed with the help of an elastic skin to adapt it to the respective flight condition. The simulations performed for this concept showed a promising increase in overall helicopter performance. The fabrication of the resulting demonstrator as well as the tests in the whirl-tower and wind tunnel are presented in this paper. The results of the tests show that the concept of linear variable chord extension has a positive influence and a great potential for hovering flight.

**Keywords:** morphing; rotor; helicopter; trailing edge; chord extension; SABRE

## 1. Introduction

Helicopters continue to be an indispensable part of aviation, as their variable speed of travel allows them to cover a wide range of operations. One important issue is the efficiency of the rotor. The design of helicopter rotor systems is always a significant compromise between the flight conditions, for example, between hover and forward flight. This results in rotor blades that are not optimal for all flight conditions, not even taking into account the different aerodynamic conditions on each rotation during forward flight. Currently rotor blade design is always a compromise between these conditions, leading to a sub-optimal performance in every condition, which increases the power required and the fuel burn beyond the theoretical possible level. Shape adaptive or "morphing" aerostructures have been a topic of significant research interest for the last decade or so, particularly in the fixed-wing community. The common goal of the diverse range of concepts investigated to date is to adapt the aerodynamic shape of an aerofoil or wing to allow for optimal performance at multiple design points. Many technological concepts for adapting the camber, thickness, planform, twist, wing tips, adaptive bumps, etc., of fixed-wing aircraft may be found in comprehensive reviews of morphing aircraft research by Barbarino et al. [1], Concilio et al. [2], and Künnecke et al. [3]. However, the applications for rotor blades are much more limited. The rotating wing poses significant additional challenges and opportunities for morphing research due to the large variety of operating conditions and highly non-linear aerodynamics, including 3D unsteady flow, compressibility, flow separation, yawed flow, and a strong influence of centrifugal loading on the blade dynamics effects as well as blade vortex interaction. On top of that, there is the large dynamic motion

of the blade in all degrees of freedom (see corresponding textbooks for details [4]). Most of this work focusses on higher harmonic excitation, which would reduce the disadvantages of the highly different aerodynamic conditions on the advancing and retreating side of the rotor in forward flight. Concepts such as discrete trailing edge flaps, "servo" flaps, as well as active twist have been proposed (see Maucher [5] for a comprehensive review) and wind tunnel tested (Boeing/DARPA/ARMY/AIR FORCE SMART HELICOPTER ROTOR [6]), and the Blue Pulse$^{TM}$ flap solution was even demonstrated by an in-flight test [7].

Much less attention was paid to concepts of quasi-static shape changes according to the flight state. Among the most successful works on this are the EU project FRIENDCOPTER (SMA based quasi-static twist) and the DARPA Mission Adaptive Rotor programs (not fully disclosed, but from what is published, includes, among others, span extension [8], deformable airfoils [9], etc.). Chord extension was intensively studied by Gandhi [10–12]), showing discrete technology bricks for both rigid body extraction as well as compliant chord extension. The rigid body extension provides a structural solution, which is robust but would need further investigation regarding aerodynamic performance with extended sheet as well as regarding dirt, icing, and noise with the opening in the trailing edge. The compliant approach solves some of these issues, but a feasible solution for the elastic skin is still to be presented. The approach in this paper will also focus on a compliant version, which works with hinged webs. Moreover, this work presents the manufacturing approach for the elastic skin, making it possible to cast the skin and bond it with webs as well as the spar. Additionally, in contrast to the state of the art, the concept shows a non-uniform chord extension, which is more of what is needed for performance increase. An advantage of this concept is the closed contour for any airfoil section without gaps and openings into the inside of the mechanism. This hinders dirt from entering the inside. The continuous surface without gaps and steps does not generate extra vortexes, which has, e.g., acoustic advances.

The desired geometry of the chord distribution can be derived from steady rotor performance considerations in hover based on combined momentum and blade element theory [4,13]. The "optimal hovering rotor" shows constant and optimum lift over drag ratio all along span, which minimizes profile power. The optimization towards constant-induced velocity distribution, which leads to minimum induced power, leads to a hyperbolic chord distribution as well as an elliptical twist distribution (see, e.g., [13]). The hyperbolic chord distribution for hover shows a longer chord length in the root area of a rotor than at the tip [14,15]. A rotor that is able to switch between a constant chord length for forward flight and a hyperbolic chord distribution for hover is a challenge for structural implementation. A first approximation of this hyperbolic distribution would be the use of two linear regions, as presented in Figure 1 [14]. This is how this morphing concept was derived. It increases the performance in hover, whereas the shorter chord length is advantageous for fast forward flight [10,15,16]. The reduction in required rotor power for this particular design at 100% extended chord over the non-extended rotor was shown to be 25 kW out of 370 kW. For an advance ratio of $\mu = 0.1$ this reduction is still shown with 14 kW out of 330 kW (see [15] for details).

Based on this assumption, a structural concept for such a chord morphing with variable blade chord length in the blade root area was developed [17], which increases the chord linearly towards the blade root starting at a pivot point. This paper presents the structural implementation of the technology into a demonstrator as well as the testing and validation in the rotor tower under centrifugal forces and in the wind tunnel.

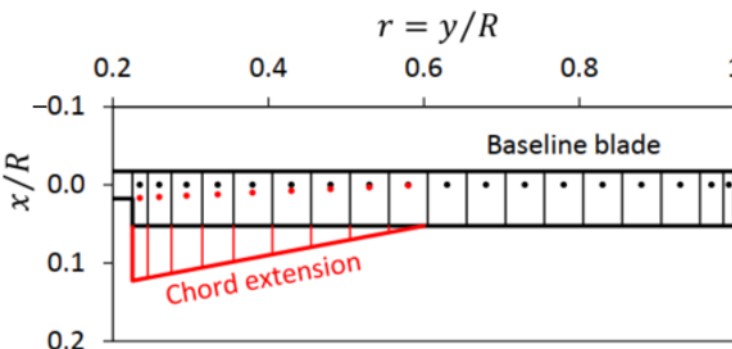

**Figure 1.** Basic principal of chord extension as a first approximation of a parabolic chord distribution (from [14]).

## 2. Concept

As the reference rotor blade is the well-documented Bo105, the structural design was shown for the NACA 23012 profile at a chord length of 270 mm. Since the radius of the test stand in the whirl-tower is limited to 2000 mm, the radius of the demonstrator is limited to 1500 mm in order to allow for circulation at the blade tip. Subtracting the rotor head and the attachment arms, this results in a total length of the demonstrator (from the attachment bolts to the tip) of 1257 mm. The morphing span is scaled in spanwise direction to fit the demonstrator size, whereby the aerodynamic area in the direction of the span is 990 mm (see Figure 2 for measurements).

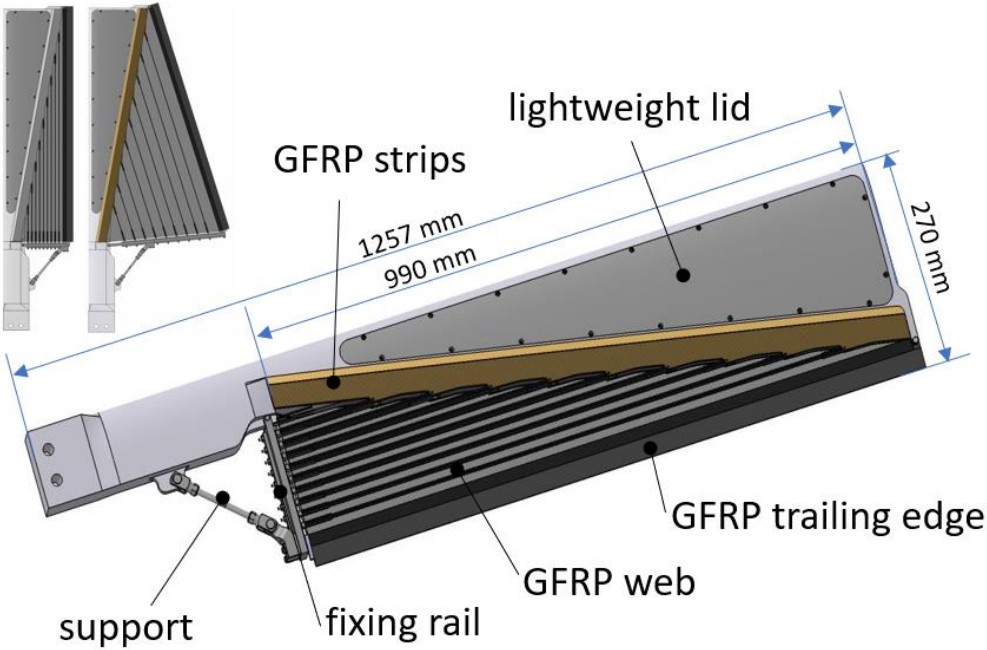

**Figure 2.** Components of the chord-morphing demonstrator and description of the individual components. Upper left corner: 0% and 100% chord morphing state of the demonstrator in whirl tower configuration.

The basis of the demonstrator is the hybrid spar, consisting of an aluminium component with glass fibre-reinforced polymer (GFRP) strips bonded to the spar for improved bonding of the EPDM. The weight-saving pocket is covered by a dedicated lightweight lid. Nine GFRP webs and a CFRP trailing edge are attached to the spar with pins. These elements form the supporting structure for the two EPDM skins on the top and bottom of the model. The skins are manufactured of type SAA9509-85 material from Gummiwerk KRAIBURG GmbH & Co KG (Waldkraiburg, Germany), which is part of the KRAIBON®

family of materials. The material was chosen due to its low Shore hardness and its high allowable strains. Moreover, it allows for a strong bonding with thermoset materials, which is the key for the material selection.

The elastic EPDM skins allow the chord length to be extended by 100% at the root area, thus increasing the surface area of the blade profile. In the future, this will be done by a dedicated actuator. As for this demonstrator, the different actuation states are realized by fixing the webs and the trailing edge to a dedicated rail. For this purpose, they are screwed to a fixing rail. In order to be able to represent discrete conditions, several fixing rails have been manufactured for this testing campaign. Rails are available for corresponding lengths of 0%, 50%, and 100% profile chord extension. This is a simplification for the test specimen. The real mechanism would have to be designed in a way that one hardware would allow for adaptation even under centrifugal loads driven with a dedicated actuation system.

To dissipate forces that occur in longitudinal direction of the blade, an additional support is provided between the fixing rail and the spar.

The test specimen has a number of requirements. The most important one is the limitation of the span of the aerodynamic surface by the wind tunnel to 990 mm. On the other hand, there are no restrictions in blade chord and thickness, so the original contour of the Bo105 with a chord length of 270 mm is used as zero actuation geometry. Since the airfoil thickness is unchanged, the airfoil itself changes during chord morphing, which is inherent to this concept and leads to different polars for different actuation states. A further limitation results from the whirl tower, where the radius from the centre of rotation to the blade tip is limited to 1600 mm, and the maximum centrifugal force of 20 kN must not be exceeded.

Due to the changed radius of the model in comparison to the original rotor blade of the Bo105, the radially outermost and innermost points of the morphing section shifts from 2.952 m and 1.081 m to only 1.6 m and 0.6 m. The main purpose of the centrifugal test is the loading and deformation of the structure and to check whether it can withstand the loads. For equivalent accelerations at root and tip of the aerodynamic morphing part, a speed of about 600 RPM [17] would be required, which serves as a reference value for the design of the demonstrator.

For manufacturing simplicity, the non-morphing parts of the spar, including the bolt attachment, are milled from aluminium. To reduce the weight of the demonstrator, a large pocket is included and covered with a lid. Since the attachment of the elastic skin to the aluminium is not possible, an auxiliary spar made from GFRP is attached (see brown part in Figure 3), which allows for bonding of the EPDM skin.

Towards the trailing edge, the individual GRP webs are positioned by tools with complex geometry, which have to be removed after vulcanization. As a further special feature, the webs are held at the radially innermost position by an articulated guide rail, which covers the entire morphing length. An additional stiffener between the guide rail and the main spar compensates the in-plane and out-of-plane forces.

The design of the individual components is primarily based on a static-mechanical FE calculation, in which the model is rotated around a virtual pivot point at a speed of 600 RPM. Additionally, aerodynamic pressure loads are applied to the model, which are based on the hover decay with maximum load. These pressure distributions are given in detail in [17].

As a result of the changes in radial location of the morphing section between helicopter and demonstrator, there is a 25% reduction in the inflow velocity of the morphing area when rotating for equal centrifugal loads. Looking at the equation for aerodynamic lift (where $c_A$ ist the lift coefficient, $\rho$ is the density of the air, $v$ is the inflow velocity, and $A$ is the cross section of the profile seen from the inflow direction), it is easy to see that the velocity enters the equation quadratically.

$$F_A = c_A \, \frac{\rho}{2} \, v^2 \, A \tag{1}$$

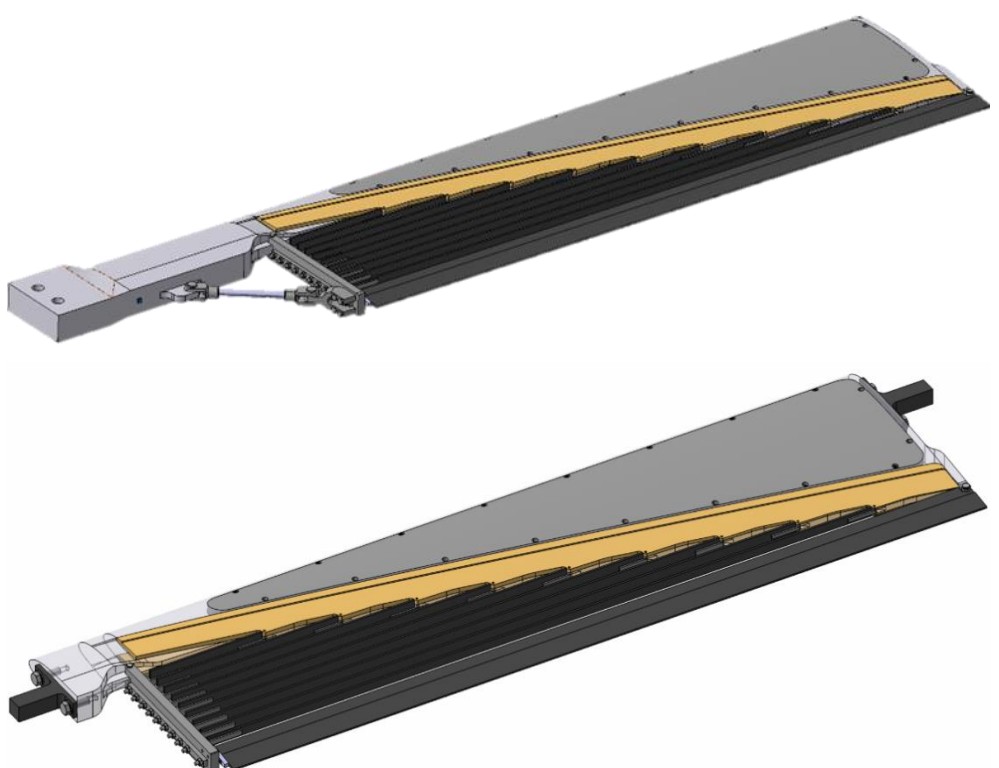

**Figure 3.** Modified demonstrator design with GFRP auxiliary spar for attachment of the elastic skin.

This almost halves the lift force and thus also the aerodynamic pressure load acting on the profile. In the following Figure 4, the deformation of the demonstrator is shown as an example:

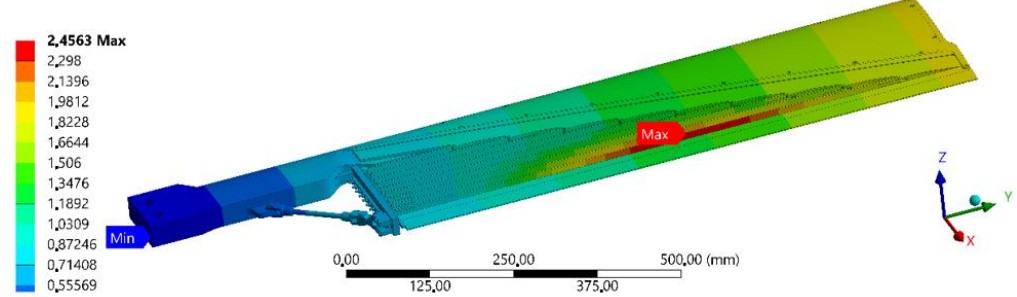

**Figure 4.** Finite element simulation (ANSYS) of the demonstrator under aerodynamic and centrifugal loads. Shown are deformations.

## 3. Manufacturing

The demonstrator consists of a combination of different materials. The aluminium spar is the basis for the entire demonstrator. Most of the components could be prefabricated or manufactured from semi-finished products. For example, the GRP inlays bonded to the aluminium spar were cut from sheet material and then milled to contour. The same applies to the CFRP trailing edge and the GRP webs, which form the support structure for the elastic skin. All these components were instrumented with strain gauges prior to any further steps. The EPDM contact surfaces of the leading and trailing edges are subsequently treated with resin and a tear-off fabric to create a defined, rough surface and improve the adhesion of the elastic skin. The fittings of the webs are glued as a separate assembly and represent the link between the aluminium spar and the fixing rail. In a final step, the elastic skin, which was produced in a calendering process, is bonded to the pre-assembled demonstrator. The temperature of 130 °C required for cross-linking,

and the ambient pressure of 4 bar make it necessary to use an autoclave. Since the EPDM temporarily loses viscosity in this process step, additional tools are required to close the space between the webs and prevent the elastic material from spreading. The contact area of the elastic skin to the webs can be increased by means of recesses filled with EPDM strips (cf. Figure 5) on the tools.

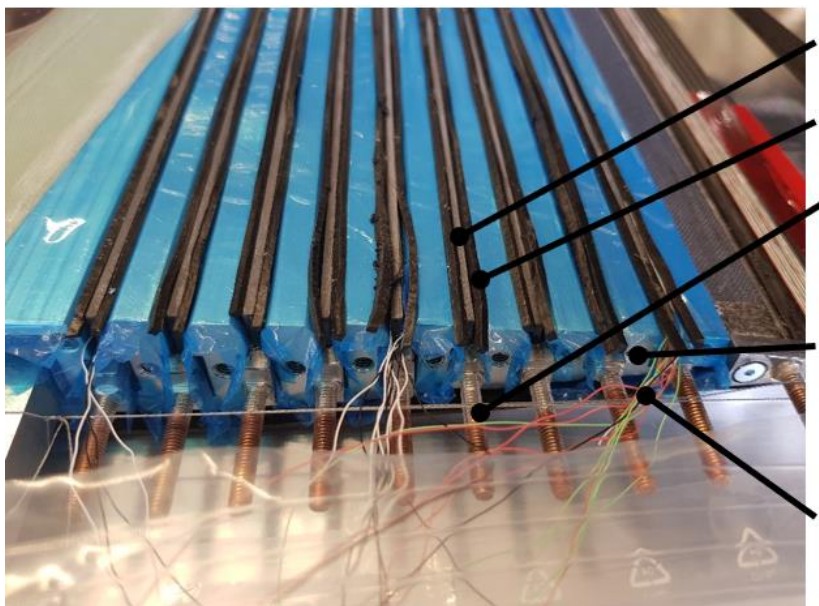

**Figure 5.** Preparation for the autoclave cycle: mould recesses are filled with EPDM strips; the upper EPDM skin is not yet applied.

After the autoclave process, the tools have to be removed again (see Figure 6). Preliminary tests have already shown that an additional separating layer is required between the tools and the elastic EPDM skin in order to overcome the adhesive bond between the two components during removal. For this purpose, the tools are wrapped in a blue release film, which is removed after removal of the tool. In addition, compressed air as a release agent has a positive effect on tool disassembly.

In a final step, the missing pocket cover of the spar is closed with a 3D-printed variant, and the demonstrator is provided with the remaining stiffening elements before it is finally installed in the test stand of the centrifugal tower (see Figure 7).

Due to a problem during the final production step, the elastic skin fell short of the nominal dimension of 1.6 mm in the root area. This results in gaps, as shown in Figure 6. Towards the tip of the blade, the skin thickness increases again. As a consequence, the skin was cut 60 mm radially from the root edge along the chord direction to avoid further tearing of the skin in radial direction. Furthermore, due to the manufacturing process, the threaded bolts of the two rear webs are missing so that they cannot be guided through the fixing rail. As a further consequence, the blade chord extension of the demonstrator was limited to 30% during the tests.

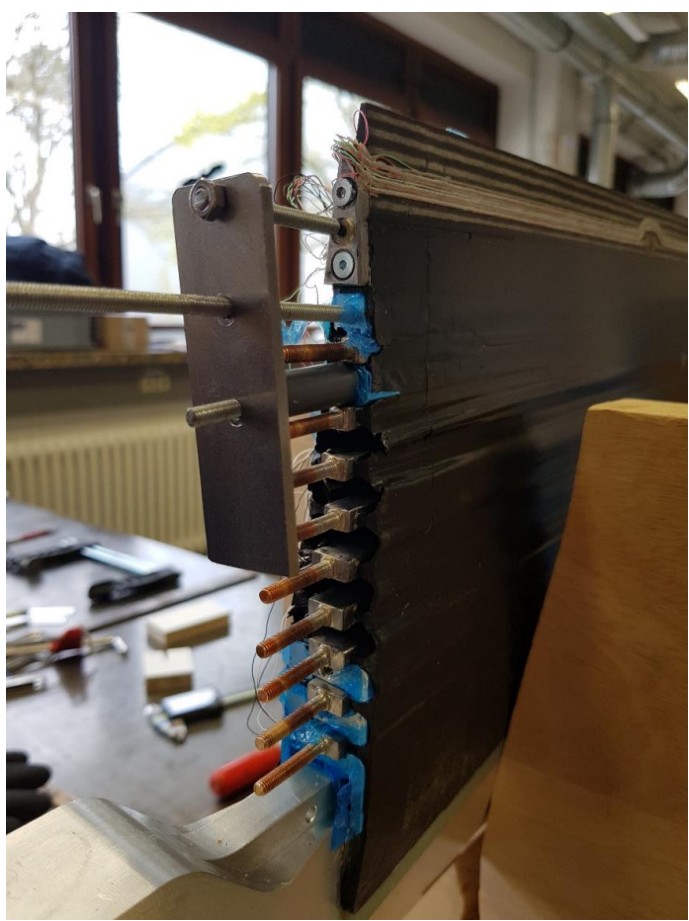

**Figure 6.** Demonstrator with tools partially removed. Pulling device is still mounted for removing the rear two tools.

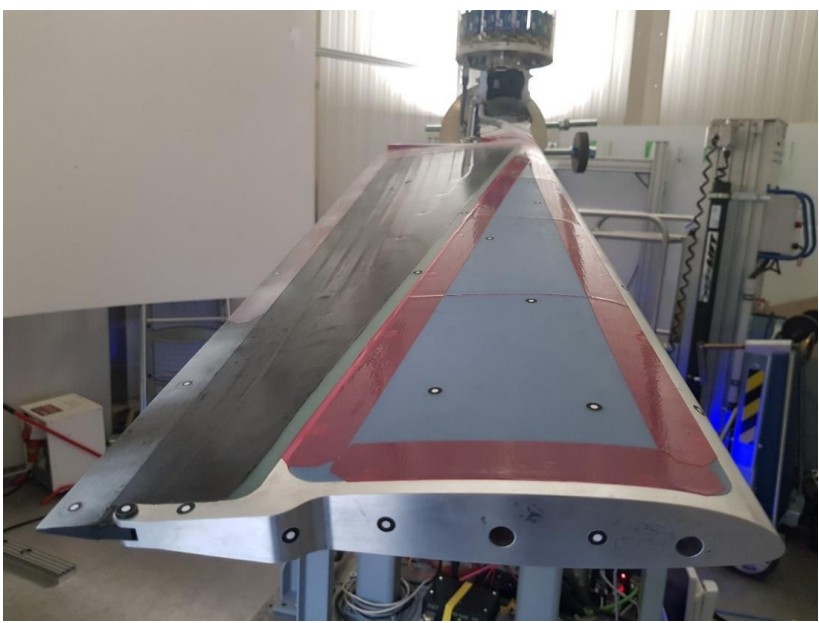

**Figure 7.** Complete demonstrator for chord morphing, installed in the test stand of the centrifugal tower.

## 4. Whirl Tower Testing

### 4.1. Strain Gauge

For the measurement in the whirl-tower, three different parameters were varied. Firstly, the chord length of the profile was tested, which assumed static values between 0% and 30% in 10% steps. Secondly, the angle of attack, which was also set statically in 5° steps between 0° and 15°, was tested. Then, thirdly, the rotor blade speed, which was increased in 100 RPM steps up to 600 RPM, was also tested. When the speed for the corresponding measuring point was reached, the measurement started for approximately 30 s. In Figure 8, the measurement results for 0% blade chord extension and 15° angle of attack are presented in detail as an example.

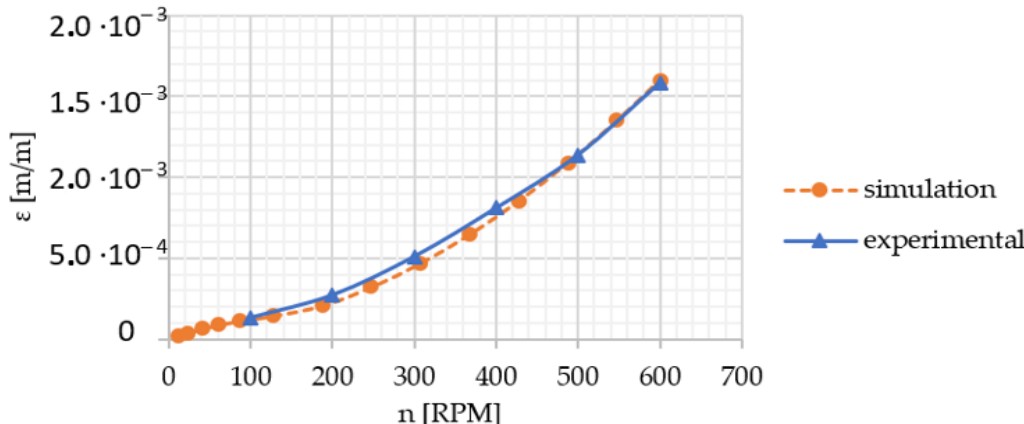

**Figure 8.** Strain curves for the strain gage at the root.

The strain gauge for the root is placed exactly in front of the hole for the fixation rail, as marked in Figure 9. All strain gauges were set to zero in the unloaded state. Strains were calculated according to their gauge factor. No additional calibration was carried out.

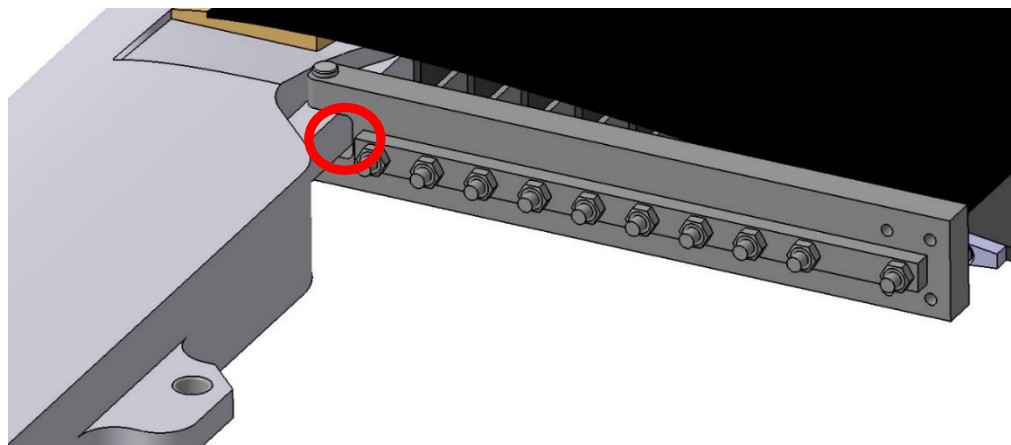

**Figure 9.** Strain gage location at the root.

Both in the simulation and in the measurement in the whirl tower, a strain increasing with speed could be measured, which speaks for a lag wise bending of the demonstrator under centrifugal loads. The degree of this sickling varies on the one hand with the speed but on the other hand also with any pretension that is induced by the root support, resulting in a delta of up to 25% in the measured strain values. The pretension has a greater effect on the strain gauge measurements of the individual webs as well as on the trailing edge itself and leads to a qualitatively similar strain curve between simulation and experiment, but quantitatively, there are corresponding discrepancies for all sensors on webs and the

trailing edge. The clamping of the webs and the trailing edge at the guiding rail as well as at the spar are the main reason for the differences between simulations and experiments. Especially, the clamping at the root region is not well captured in the model.

### 4.2. Surface Accuracy of the Skin

Another aspect of the experimental phase is to check the accuracy of the outer contour between the demonstrator and the FE simulation for different morphing states. The Atos system was used for this purpose. This is an optical 3D scanner based on fringe projection that provides traceable 3D coordinates. The results of the surface analysis are shown below for the 30% chord extensions.

The spar serves as a reference for the alignment of the two surfaces to each other. As can be seen from the top three measuring points at the leading edge in Figure 10, the geometries from the FE and Atos measurements fit very well.

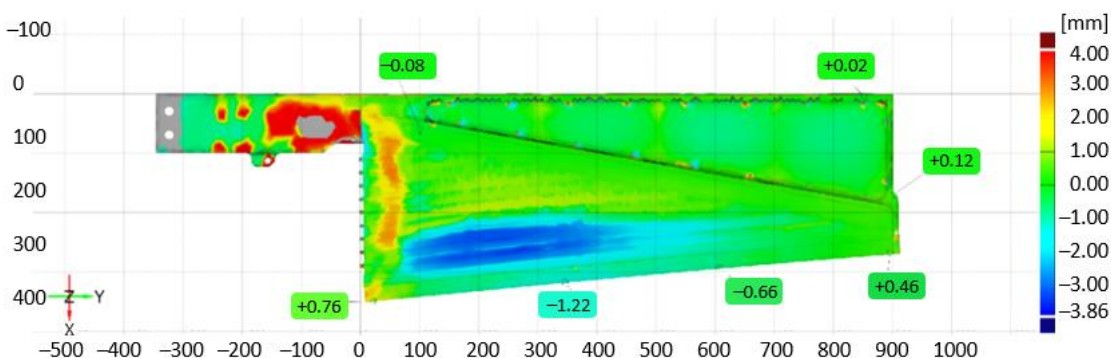

**Figure 10.** Difference of outer geometry between simulation and experiment for 30% chord morphing. Positive values stand for measured surfaces higher than the simulation; negative values indicate a measurement that is below the expectation of the simulation.

The larger deviations in the root area of the spar result from the cabling, which was additionally fixed with adhesive tape. In the root area of the skin, the deviations are mainly due to the pre-damage of the skin caused by the production process. Significant deviations are also found in the area of the last web, which is located directly in front of the trailing edge. As described previously, no guidance or defined bearing is possible here due to the missing threaded welding stud.

The measuring points at the trailing edge show an out-of-plane deformation in the positive z-direction, which can be explained by the minimal clearance in the joints of the trailing edge with the spar at the blade tip as well as the fixing rail with the spar in the root area, which are less tolerated in the FE calculation. In addition, the trailing edge deforms centrally in the negative z-direction due to the tensile forces resulting from the blade chord extension. At this point, the simulation seems to show higher stiffnesses than reality. The hyperbolic course of the trailing edge can also be seen.

Figure 11 shows a sectional view along the blade profile at 100-mm radial distance from the EPDM edge of the root area. The surface of the demonstrator captured by the Atos system for the entire blade profile is shown. The colours and markings here correspond to the geometric deviation in the z-direction of the demonstrator from the FE simulation in millimetres. It can be seen that the position of the first five webs corresponds very well with the FE simulation. This is no longer the case for the last four webs. This is understandable because the last two webs are no longer held in position by the fixation rail, and therefore, the force transmission only takes place via the elastic skin. This also reduces the tensile force on the two subsequent webs, which can deform accordingly 100 mm from their fixed clamping on the fixation rail.

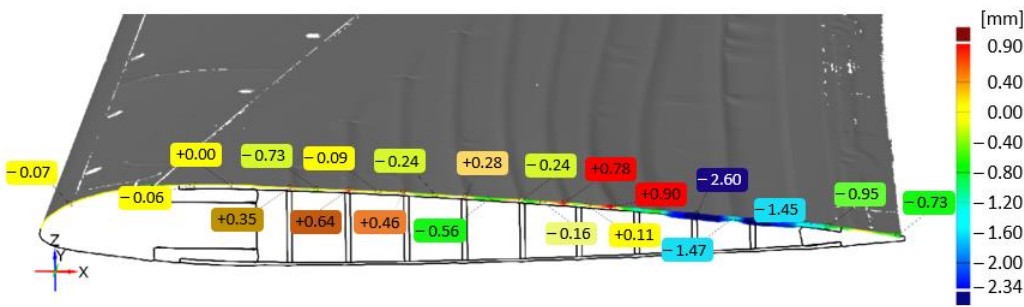

**Figure 11.** Surface comparison between simulation and experiment in a sectional view, 100 mm from the EPDM edge. Positive values stand for measured surfaces higher than the simulation; negative values indicate a measurement that is below the expectation of the simulation.

　　　Furthermore, an offset or downset of the surface can be observed in the area of the first five webs. The main reason for this will be installation tolerances. Slight deviations due to the asymmetrical profile and the associated expansion differences between the upper and lower side are also possible as well as variations in the skin thickness due to manufacturing. Of greater interest, however, is the relative difference in height between the areas bounded by two webs and the adjacent webs themselves. These are relatively close: 0.75 mm between the first and second webs and 0.58 mm between the fourth and fifth webs. From this, it can be deduced that the real out-of-plane deformation in the z-direction is greater than originally assumed. The reason for this may be the accumulation of material in the area of the webs, which are supposed to form a better connection, and the associated increased stiffness. This increases the absolute elongation between the webs and the resulting buckling. For a detailed representation, Figure 12 shows the section courses from Figure 11 once again with unequal axes. Note an offset of 1 mm in the simulation data (red pluses) for better clarity. This illustrates the differences and similarities between simulation and measurement already described.

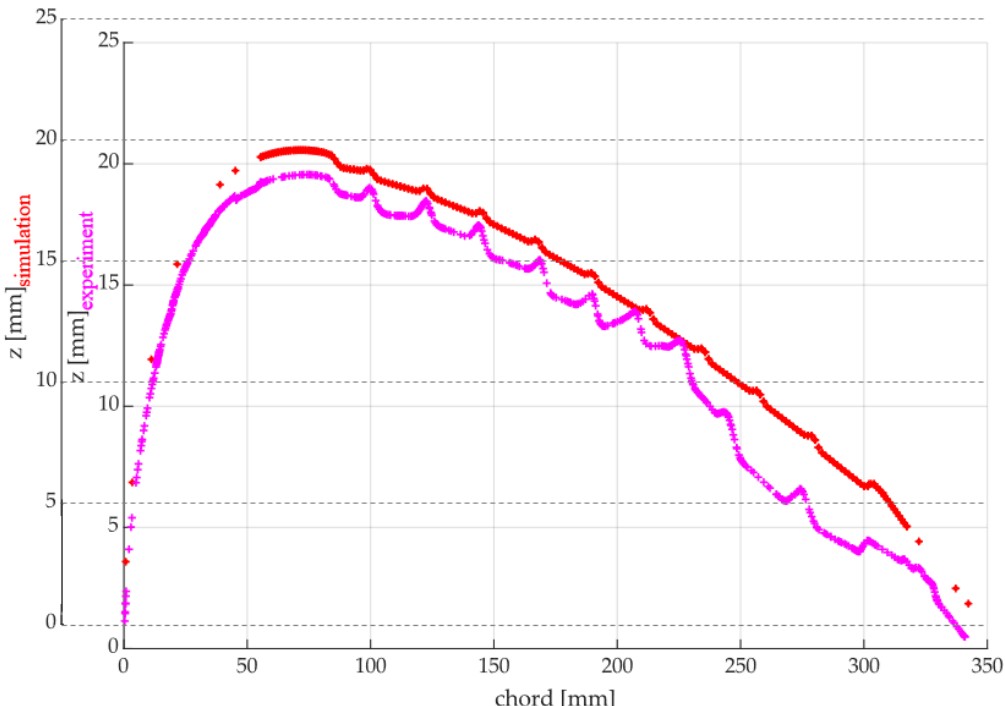

**Figure 12.** Comparison of the surfaces of Atos measurement (pink plus) and simulation (red pluses) in a sectional view with unequal axes. Values of the simulation have an offset of 1 mm for better clarity.

## 5. Wind Tunnel Testing

### 5.1. Equipment and 2D Insert

A quasi-2D wind tunnel experiment was performed to assess the aerodynamic performance of the variable chord extension wing presented in this article. This experiment was conducted in the University of Bristol's 7 ft × 5 ft low-speed wind tunnel.

Since the University of Bristol's 7 ft × 5 ft low-speed wind tunnel was not previously set up to perform two-dimensional wind tunnel experiments, the facility had to be adapted. A 2D wind tunnel insert was designed and built in-house and was equipped with off-the-shelf instruments and sensors to measure the forces and moments of the different wing models.

A bespoke 2D wind tunnel insert with a 1-m × 1-m cross-section by 1.2 m-long test section was designed and built. The 2D insert consists of two splitter plates made of 10-mm-thick acrylic sheets mounted on square frames and four vertical uprights connecting them and mounting the wing section. This frame is made from extruded aluminium slotted rail sections for ease of assembly and modification. The uprights support the top splitter plate and maintain the 1-m section height. Moreover, the connections at all corners are made of four distinct brackets each to ensure high rigidity and strength (Figure 13). The bottom splitter plate is mounted to a base frame, where the automatic pitch drive system also sits, through four shorter vertical extrusions that lift the cube-like structure to the centre of the tunnel section (Figure 14). The splitter plates work as walls for the reduced test section where 3D effects are reduced, and a desired quasi-2D flow is produced on a 1-m span. The leading and trailing edges of the splitter plates were cut at a 45° angle to cleanly cleave off a fresh boundary layer. The two sides, which correspond to the walls on the suction and pressure side of the flow, are open, allowing the free expansion of the freestream along the airfoil thickness. An additional measure to mitigate the presence of 3D effects was monitoring the freestream velocity at three different locations using three independent Pitot-static tubes so that variations in speed could be accounted during the data analysis step.

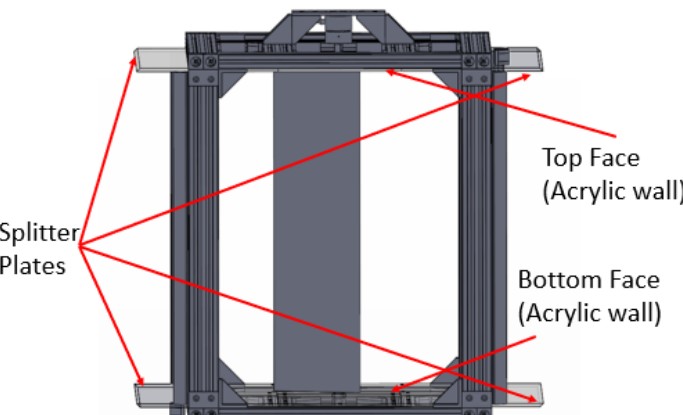

**Figure 13.** 2D test section: acrylic walls and splitter plates.

A precision right-angled worm gearbox driven by a synchronous servo motor was installed with the axis in a vertical orientation and used to control the wing's pitch angle. An industrial digital drive with 6 A-rated output was installed to drive the servomotor and set up with closed-loop control of position and analogue output to be acquired during testing. The servo produces a stall torque of 2.9 Nm with a resolver measurement accuracy ±45″ and is equipped with a braking system. The two-stage worm gearbox has a reduction ratio of 400:1 and can support a maximum axial load of 25 kg. The system was designed to produce a maximum pitching torque of 100 Nm and operate between 0.1 and 12 rpm, with an overall accuracy of 0.0667° (driven mostly by gearbox backlash). The dedicated

software has been set up to perform sequential tasks corresponding to the angle ranges and step increases defined for each test.

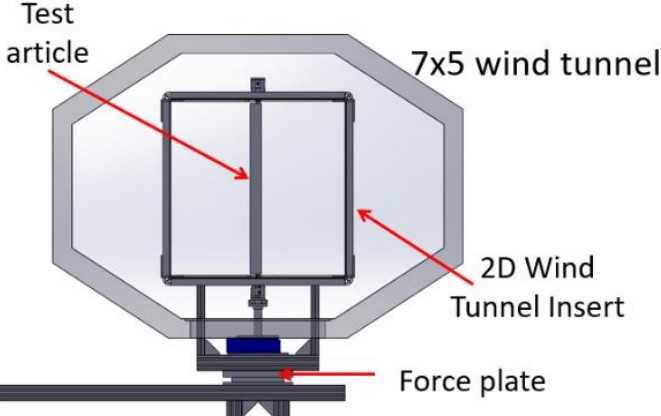

**Figure 14.** 2D wind tunnel insert inside 7-ft $\times$ 5-ft test section.

Two ME-systems K6D80 six-axis force sensors with a forces and moments range of 2 kN and 100 Nm, respectively, were used to measure the aerodynamic forces and moments acting on the wing. Two load cells are required to fix the demonstrator at both ends and obtain 2D aerodynamic coefficients. The top force sensor was mounted to the 2D insert using a series of aluminium brackets and is integral with it, whereas the bottom force sensors were attached to a shaft connected to the pitch drive that rotates inside a flanged bearing to change the wing's pitch angle. Mounting and attaching the wing directly to both force sensors allows bypassing the aerodynamic forces acting on the 2D insert (e.g., additional drag). However, this setup does not eliminate the effect that the presence of the uprights may have in terms of flow quality and smoothness.

Furthermore, the entire 2D insert structure is mounted to an AMTI Biomechanical Force Platform (1000 lb max measurable force), which measures the forces and moments that the entire setup experiences during the wind tunnel test. The force plate is fixed to a foundation frame, which connects to the wind tunnel structure. These forces and moments are measured as a "backup" to the wing's force sensors. Therefore, this data set would only be used if the force sensors present any issues. Additionally, two pitot tubes were mounted to the two front uprights (i.e., facing the inlet of the test section) to measure the local flow velocity where the quasi-2D flow is produced. Each pitot tube is connected to one NXP MPXV7002 differential pressure sensor, which measures the dynamic pressure. Due to blockage (around 10% for the 2D insert plus wing), the flow velocity inside the 2D test section is around 10% higher than the wind tunnel's speed (measured upstream the wind tunnel test section). Therefore, the local freestream velocity inside the 2D insert is used for all aerodynamic coefficient calculations. Moreover, two thermocouples were placed on each pitot tube to record temperature during testing. Lastly, an NXP MPXA6115AC6U 115 kPa Absolute Pressure Sensor was used to record the atmospheric pressure. This measured pressure, along with the temperature measurements, was then used to calculate the air density. Lastly, for safety purposes, two accelerometers were installed to the 2D insert to monitor vibrations.

The selection of wind speeds for this setup was limited to technical considerations. Due to vibrations at the setup (measured by the above-mentioned accelerometers) at speeds of 35 m/s and above, it was decided to stay under that limit. Freestream velocities measured at low speeds, such as 10 m/s and below, suggested that 3D aerodynamic effects dominated the flow profile due to continuous observed oscillations on the velocities measured by the three Pitot tubes. Since the authors wanted to test at two different Reynolds numbers, it was decided to test at 20 m/s and 30 m/s.

### 5.2. Wind Tunnel Test

As the results from the wind tunnel tests show, the rotor blade chord extension has a positive influence on the rotor's characteristics. The results of the wind tunnel tests at a speed of 20 m/s and 30 m/s are shown in Figure 15. In total, the three morphing states with 0%, 15%, and 30% chord extension were investigated at angles of attack between $-5°$ and 15°. It is worth mentioning that, since (a) this wind tunnel setup consists of a 2D wind tunnel insert inside the $7' \times 5'$ test section and (b) no local 2D drag-measuring technique was implemented (e.g., wake rake), the drag measurements are expected to be higher and more three-dimensional than if a 2D drag-measurement technique was implemented. Therefore, these drag estimates should be considered qualitatively. To correct for three-dimensional effects, standard wind tunnel corrections are applied to account for solid blockage and streamline curvature. Because the 2D insert is a closed section with no top/bottom covers (i.e., both suction and pressure sides are left open), the wake blockage is assumed to be zero. The correction factors implemented in this study are the classical ones given by [18].

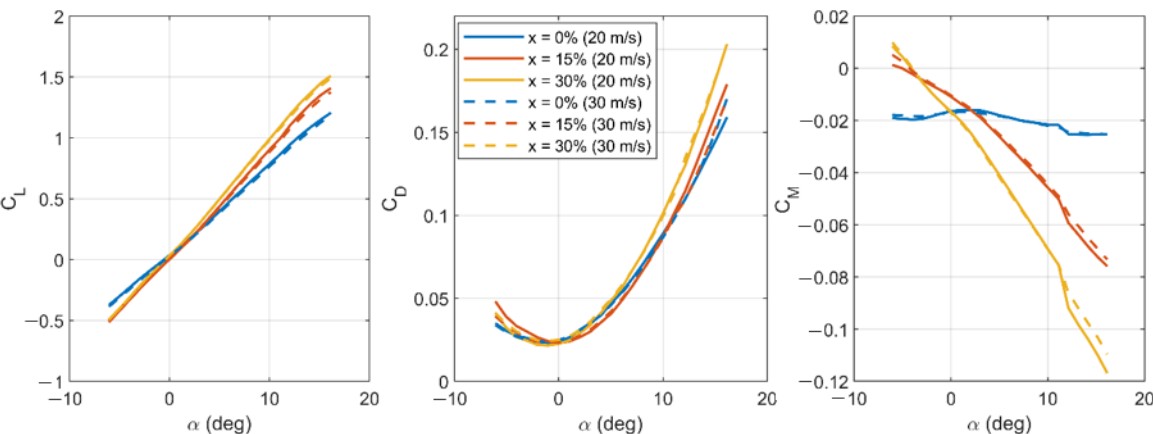

**Figure 15.** Results for the lift (CL), drag (CD), and pitching moment (CM) coefficients of the wind tunnel test at 20 m/s and 30 m/s with an angle of attack sweep from $-5°$ to 15°.

Most importantly, increasing the root chord is shown to be an effective means of increasing lift. Note that the results here use the baseline chord in order to calculate the lift coefficient, as is standard practice for things like fowler flaps, which also increase chord. Therefore, the increase in lift coefficient seen with chord extension is caused first and foremost by the increased chord, with other effects possibly included. This materializes as essentially an increase in lift curve slope of the wing section. If we consider a change from 0% to 30% morphing, the lift curve increases by a remarkable $\Delta CL\alpha$ = 29%. This is even more significant when we consider that the entire span of the wing does not increase in chord; indeed, the average increase in chord over the wing is on the order of 15%. This alludes to some more complex flow physics taking place, perhaps including some amount of camber change or an increase in lift due to the increased skin tension. It is also likely that there is some impact from the test rig itself, such as interference effects within the 2D insert, specifically because the increase in chord also increases in blockage in the insert leading to global changes in the flow environment within the tunnel that may produce higher effective angles of attack.

We can also see that the changes in drag due to morphing are measurable but less significant than the changes in lift. Due to the presence of a large fairing on the base of the model and the not perfectly two-dimensional flow in the insert, the exact values of drag do not have much physical meaning; instead, it is the variation in drag with morphing that we consider.

While the more detailed aspects of both the lift and drag results are clearly in need of further investigation in future research, the measured changes in lift curve slope

are very significant and demonstrate the underlying aerodynamic efficacy of the chord change mechanism.

## 6. Conclusions

Based on the tests in the whirl tower and wind tunnel, a successful implementation of the structural concept for chord morphing could be demonstrated, which is the very first complete demonstration of this novel chord-morphing concept with linear chord distribution. It could be shown that the KRAIBURG EPDM is suitable for the production of elastic skins, and a manufacturing strategy was presented, which enables the adhesion to spar, webs, and trailing edge in a single process. This successful process is a milestone for the manufacturing of flexible skins and the implementation of the needed supports. During the tests, the deformations of such skins under different loads was measured and presented. It was shown that just the stretching itself will lead to some contour deviation due to the interaction with the supporting webs. The order of magnitude of such changes have been demonstrated and will be a critical source for future works on elastic skins. Additionally, the sickling deformation of the blade was shown in under centrifugal loads. The interface of the webs at the root were identified as sources of differences between the simulation and the experimental strains in the overall system. Finally, the wind tunnel tests have shown the expected influence of the chord length on the aerodynamic coefficients.

To sum up: besides the demonstration of these concepts' general feasibility, many technical details need attention in the future. To push this technology to a higher TRL, work has to be done in the areas of overall blade design with respect to the spars dimension, connection of moving webs to fixing rail for load transfer, dimensioning of the EPDM, and the sub structure to reach sufficient surface smoothness and an actuation concept.

**Author Contributions:** Conceptualization, C.B. and J.R. formal analysis, C.B.; investigation, C.B., S.K., J.R. and A.R.; writing—original draft preparation, C.B. and A.R.; writing—review and editing, C.B., J.R. and A.R.; visualization, C.B. and S.K.; supervision, J.R.; project administration, J.R.; funding acquisition, J.R. All authors have read and agreed to the published version of the manuscript.

**Funding:** This research was funded by the European Commission grant number 723491–SABRE project.

**Data Availability Statement:** The data presented in this study are available on request from the corresponding author.

**Acknowledgments:** My special thanks go to the colleagues at DLR Braunschweig who made the successful realization of the project possible but also to the colleagues from Bristol for the excellent cooperation. In addition, many thanks to the company Gummiwerk KRAIBURG GmbH & Co KG, which provided the material for the elastic skin, and for the excellent discussion including some tips for the materials processing.

**Conflicts of Interest:** The authors declare no conflict of interest.

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
