# Peer review of "Manufacturing and Testing of a Variable Chord Extension for Helicopter Rotor Blades"

_actuators, doi:10.3390/act11020053_

Round 1
Reviewer 1 Report
In general, please try to improve the text for clarity. Also, Introduction must be expanded, there are many other morphing blade concepts (telescoping wing) or means of improving aerodynamic performance. Please see the attached file for specific comments.

Author Response
Titel has been changed according to request
Citations have been removed from abstract
Introduction has been extended. State of the art has been improved. Dimensions have been added to the figure 1.
Numbers in figure 1 were substituted by labels.
More clarity for the mentioned differences in strains and contur deviations were given in thext.
Small comments were considered and changes were included.
Wind tunnel section:
- Why these wind speeds
Answer: even though the wind tunnel has a maximum test speed of 50 m/s, the test operators found out that excessive vibrations occurred when the test speed exceeded 35 m/s. These vibrations were possibly due to structural dynamic interactions between the air flow and the 2D wind tunnel insert. Thus, it was decided not to exceed 35 m/s to minimise vibrations. On the other hand, when operating below 15 m/s, the three freestream velocity measurements showed inconsistent readings. This inconsistency suggests that 3D aerodynamic effects dominated the flow profile when below 10 m/s. Thus, it was decided to test above 15 m/s. Since the authors wanted to test at 2 different Reynolds numbers, it was decided to test at 20 m/s and 30 m/s.
An additional paragraph was added
- Tunnel dimensions in none metric
Answer: it i true that the wind tunnel dimensions ar non-metric. However, the authors decided to leave them in ft. because a 7 ft x 5 ft is a traditional wind tunnel size and researchers in the experimental aerodynamics community refer to these type of wind tunnels as "7x5 tunnels". Thus, the authors consider that it is appropriate to keep the dimensions in ft for historical reasons.
- Three result pictures in one
Answer: In the current graph we can see the differences that we get from the influencing factors. The authors believe, that this is an important learning. If it would be split apart, this expected effect would not be shown as clear.
Reviewer 2 Report
In this paper, the authors report the aerodynamic results of the model chord-morphing rotor. It seems that they successfully developed the variable chord rotor model, tested, and got some reasonable data from the tests - the chord extension increased the overall lift coefficient. What could be the new findings or the novelty of this paper, however, is not clear. The introduction which should provide sufficient research background is too weak. More than several unclear expressions that should be revised are found.
1. L11: “A particularly important role is played by …” Please avoid such a not informative and indirect expression as much as possible. Revise it in a concise manner.
2. L15 & L17: Please check the rule of MDPI; I think most journals do not allow the use of citations in Abstract unless it is critical for the contexts. Also, consider the order of numbers
3. L24-L34: These two paragraphs are the part of the introduction (the paragraph from L35-L43 should move to the Material and Method). Too weak and unkind. The authors should give a chance to the potential readers to imagine the same research background. Much more details are quite necessary. Please consider: why this research need? Which studies have been performed for a similar goal? How about their study? What was missing? What does this study fill out? At least a one-page description needs at the minimum.
4. L36: “Special attention was..” looks redundant.
5. L66: “…so the original…” This means the model eventually has a much shorter spanwise length with a smaller aspect ratio. Was there no further aerodynamic effect? Can you explain or defend this question of “the model just loses the geometrical similarity.”?
6. L93: What is this description mean? Can you elucidate it?
7. L104: “mass and volume” looks awkward.
8. L146: What is this mean? sub-title for the paragraph? this is not clear; make it with full sentences.
9. L158: “… the illustration 8” In usual, the Figure and Table are treated as proper nouns in a scientific paper. This is to secure concise description and scientific clearness. Check this and other terminologies and revise in a concise manner.
10. L164: “… pretension…” What is this mean? If the authors denoted the pretension in straingauge then, I should say, it is completely removable, by using the “delta” strain from the reference (no load).
11. L168: “the following diagram”, which also should be revised as Figure 9.
12. L199: Which one is the simulation and which is the measurement in Figure 10?
13. L232: “Values of the simulation has an offset of 1 mm for better clarity”. This is worse and very dangerous because it can be rather misunderstood by the reader! Please remove the offset and use a different color, line, thickness to give proper visibility.
14. L259: This reviewer would like to recommend exchanging the order of Figures 13 and 14, because Figure 14 shows the overall configuration and is easier to imagine. Related descriptions should also be exchanged.
15. L345: Conclusion should be standalone as “an independent paragraph”. It should be readable without any other content. Revise the conclusion following this manner.
Author Response
In this paper, the authors report the aerodynamic results of the model chord-morphing rotor. It seems that they successfully developed the variable chord rotor model, tested, and got some reasonable data from the tests - the chord extension increased the overall lift coefficient. What could be the new findings or the novelty of this paper, however, is not clear.
The introduction which should provide sufficient research background is too weak. Introduction was significantly improved. State of the art is presented in more detail and the reasoning for the particular research is given in comparison to the state of the art.
More than several unclear expressions that should be revised are found.
- L11: “A particularly important role is played by …” Please avoid such a not informative and indirect expression as much as possible. Revise it in a concise manner. Whole paragraph revised
2. L15 & L17: Please check the rule of MDPI; I think most journals do not allow the use of citations in Abstract unless it is critical for the contexts. Also, consider the order of numbers Citation in abstract erased
3. L24-L34: These two paragraphs are the part of the introduction (the paragraph from L35-L43 should move to the Material and Method). Too weak and unkind. Paragraph moved out of introduction. The authors should give a chance to the potential readers to imagine the same research background. Much more details are quite necessary. Please consider: why this research need? Which studies have been performed for a similar goal? How about their study? What was missing? What does this study fill out? At least a one-page description needs at the minimum. Added
4. L36: “Special attention was..” looks redundant. Section completely rephrased
5. L66: “…so the original…” This means the model eventually has a much shorter spanwise length with a smaller aspect ratio. Was there no further aerodynamic effect? Can you explain or defend this question of “the model just loses the geometrical similarity.”? Scaling explained more in detail.
6. L93: What is this description mean? Can you elucidate it? Explained more clearly
7. L104: “mass and volume” looks awkward. removed
8. L146: What is this mean? sub-title for the paragraph? this is not clear; make it with full sentences. changed
9. L158: “… the illustration 8” In usual, the Figure and Table are treated as proper nouns in a scientific paper. This is to secure concise description and scientific clearness. Check this and other terminologies and revise in a concise manner. changed
10. L164: “… pretension…” What is this mean? If the authors denoted the pretension in straingauge then, I should say, it is completely removable, by using the “delta” strain from the reference (no load). changed
11. L168: “the following diagram”, which also should be revised as Figure 9. changed
12. L199: Which one is the simulation and which is the measurement in Figure 10? Changed. The delta is shown
13. L232: “Values of the simulation has an offset of 1 mm for better clarity”. This is worse and very dangerous because it can be rather misunderstood by the reader! Please remove the offset and use a different color, line, thickness to give proper visibility. updated
14. L259: This reviewer would like to recommend exchanging the order of Figures 13 and 14, because Figure 14 shows the overall configuration and is easier to imagine. Related descriptions should also be exchanged. switched
15. L345: Conclusion should be standalone as “an independent paragraph”. It should be readable without any other content. Revise the conclusion following this manner. rewritten
Reviewer 3 Report
The paper considers the effect of linear variable chord extension in a helicopter blade on the different flight parameters via experimental and simulation studies. The work is interesting and worth studying; however, there are several issues to be considered in the current format, as follows:
- The reference orders must be corrected in the manuscript, for example, reference [1] appears after reference [4] in Introduction.
- The labels should include in all the figures, for instance in Figure 12 it is missed. Also, it indicates in the caption that blue line while there is not a blue line in the figure.
- More information on the accelerometers and the way they are mounted should be included in the manuscript.
- Why component number 3 is not shown in Figure 1.
- It should be explained how the strain gauges are calibrated for the tests.
- The sentences should be more informative and complete, an example is in Conclusion:
“Nevertheless, the test data, especially from the wind tunnel tests, have shown that they have great potential.”. It should be described what are the potentials?
- There are several typo and grammatical mistakes in the manuscript that must be corrected, such as:
Subtitles are started with the small letter “strain gauge”.
“at 20 m/s and 30m/s with”
“two-dimensional flow”
“This is the very first complete concept demonstrator of this novel morphing concept.”
Author Response
The paper considers the effect of linear variable chord extension in a helicopter blade on the different flight parameters via experimental and simulation studies. The work is interesting and worth studying; however, there are several issues to be considered in the current format, as follows:
- The reference orders must be corrected in the manuscript, for example, reference [1] appears after reference [4] in Introduction. References rearranged and extended
- The labels should include in all the figures, for instance in Figure 12 it is missed. updated Also, it indicates in the caption that blue line while there is not a blue line in the figure. updated
- More information on the accelerometers and the way they are mounted should be included in the manuscript. Since the accelerometers do not have any meaning for the scientific aspects of the paper, but only for the safety of the tunnel and to observe possible instabilities of the setup in the tunnel, the authors choose not to give more detailed about this system. The meaning of the vibrations is elaborated on in the following passage.
- Why component number 3 is not shown in Figure 1. Numbers replaced by descriptions
- It should be explained how the strain gauges are calibrated for the tests. implemented
- The sentences should be more informative and complete, an example is in Conclusion:
“Nevertheless, the test data, especially from the wind tunnel tests, have shown that they have great potential.”. It should be described what are the potentials? Section rewritten
- There are several typo and grammatical mistakes in the manuscript that must be corrected, such as:
Subtitles are started with the small letter “strain gauge”.
“at 20 m/s and 30m/s with” changed
“two-dimensional flow”
“This is the very first complete concept demonstrator of this novel morphing concept.” changed
Round 2
Reviewer 1 Report
There is substantial work in the paper however it lacks clarity, which reduces its soundness. I recommend a thorough reading and have substantial editing in the text. Some pictures also need improvements, which are difficult to follow. I also have some notes in the attached document.

Author Response
43: wording of the additional challenges of rotating wings was expanded and a citation was added.
62: The advantages are on one hand side a closed contour in chord vise direction (no dirt, acoustic issues): text was adapted.
67: description of theoretical backbone updated
108: figure of fully closed and fully open configuration added
126: the weight reduction is only meant for the demonstrator. Due to the different materials used, this cannot be compared to the composite built BO 105 blade. The real design would not be much increased than the reference. This has to do with the low weight of the elastic skin as well as the loss of the foam in the inside. This compensates for the webs. However, the guiding rail, that holds the webs is not considered in this weight estimation at this stage in the development, but would be significant.
144: the reduction comes from two effects. 1: the radial location of the morphing part differs between helicopter model and demonstrator, due to different rotor hub geometries, attachment radius… second the rotational speed for the test was chosen according to centrifugal load considerations in the morphing section, which differ from the original design, since the radial distance from the rotational hub has changed. These two effects lead to a reduction in rotational speed in the particular region.
153: software and mention in text
228: The reviewer’s comment is valid. For this reason the figure will be removed.
258: added explanation to the caption
287: added explanation to the caption. Regarding the size of the numbers: the authors can only process, what the digitizer software gives them. So the authors tried to enlarge the overall picture as much as possible. In our opinion it still is not perfect, but just acceptable. The alternative would be to leave out this plot, which the authors try to avoid.
321: You wanted to conduct 2D tests as understood. However, the blade seems to be located very close to the inlet of the 2D insert. The flow separation at the insert may cause 3D effects. Please explain.
It is true that some 3D effects were present inside the test section. However, that is why the authors refer to a "quasi-2D" test condition. The authors are well aware of the limitations of the setup.
Several measures were introduced to mitigate the presence of 3D effects, such as installing splitter plates at both inlet and outlet, keeping the side walls of the insert uncovered so that air can freely expand along the through-thickness direction of the wing, and monitoring the freestream velocity at three different locations inside the tunnel (independent pitot-static tubes).
366: Why: Freestream velocities measured at low speeds such as 10 m/s and below suggested that 3D aerodynamic effects dominated the flow profile, due to continuous observed oscillations on the velocities measured by the three Pitot tubes.
Reviewer 2 Report
accept as is
Author Response
Thanks for accepting.
Reviewer 3 Report
The article could be publised.
Author Response
Thanks for accepting.